# BNIP3 in Lung Cancer: To Kill or Rescue?

**DOI:** 10.3390/cancers12113390

**Published:** 2020-11-16

**Authors:** Anna S. Gorbunova, Maria A. Yapryntseva, Tatiana V. Denisenko, Boris Zhivotovsky

**Affiliations:** 1Faculty of Basic Medicine, Lomonosov Moscow State University, 119192 Moscow, Russia; gorbunovaanna94@gmail.com (A.S.G.); smariaal@mail.ru (M.A.Y.); de_tanya@yahoo.com (T.V.D.); 2Karolinska Institutet, Institute of Environmental Medicine, SE-17177 Stockholm, Sweden

**Keywords:** BNIP3, lung cancer, apoptosis, autophagy, metastasis

## Abstract

**Simple Summary:**

Bcl-2/adenovirus E1B 19kDa interacting protein 3 (BNIP3) is a pro-apoptotic BH3-only protein of the Bcl-2 family. Its function in various biological processes was described. Although potential involvement of BNIP3 in cancer progression has been discussed in many review articles, its specific role in lung cancer is still unclear. In this review, we shed light on the BNIP3‘s role in different types of cancer in general and lung cancer, in particular, as well as suggested its potential for targeting therapy of lung cancer.

**Abstract:**

Bcl-2/adenovirus E1B 19kDa interacting protein 3 (BNIP3) is a pro-apoptotic BH3-only protein of the Bcl-2 family. Initially, BNIP3 was described as one of the mediators of hypoxia-induced apoptotic cell death in cardiac myocytes and neurons. Besides apoptosis, BNIP3 plays a crucial role in autophagy, metabolic pathways, and metastasis-related processes in different tumor types. Lung cancer is one of the most aggressive types of cancer, which is often diagnosed at an advanced stage. Therefore, there is still urgent demand for reliable biochemical markers for lung cancer and its efficient treatment. Mitochondria functioning and mitochondrial proteins, including BNIP3, have a strong impact on lung cancer development and progression. Here, we summarized current knowledge about the BNIP3 gene and protein features and their role in cancer progression, especially in lung cancer in order to develop new therapeutic approaches associated with BNIP3.

## 1. Introduction

According to the World Health Organization, lung cancer is the leading cause of cancer mortality worldwide: more than 1.5 million people die from this disease every year. Lung cancer is histologically divided into small-cell lung cancer (SCLC) and non-small-cell lung cancer (NSCLC). The number of NSCLC cases accounts for 80–85% of all lung cancers and, accordingly, prevails over SCLC, the incidence of which is 10–15%. NSCLC is subdivided into three big subtypes, namely, squamous cell carcinoma (SCC), large-cell carcinoma (LCC) and adenocarcinoma (AD) [1]. Cigarette smoking is related to all subtypes of lung cancers, both in smokers and in second-hand smokers [2]. Besides smoking, the development of lung cancer is associated with genetic predisposition and environmental exposure [3]. Lung cancer is linked with somatic mutations in the *EGFR*, *TP53*, *KRAS*, and *ALK* genes [4,5]. Concomitant *TP53* mutation in *EGFR*-mutated NSCLC patients is associated with poor prognosis [6]. Another lung cancer feature is a high ability to invade neighboring tissues and metastasize to distant organs. There is a growing body of evidence that mitochondria contribute to cancer development and progression, including lung cancer [7]. Bcl-2/adenovirus E1B 19kDa interacting protein 3 (BNIP3) is a mitochondrial BH3-only member of the Bcl-2 protein family. BNIP3 is activated by hypoxia and mainly functions as a cell death regulator in hypoxic conditions [8]. It affects different cell death pathways, including apoptosis, necrosis-like cell death, autophagy, and its special form, mitophagy. These processes are also involved in epithelial-mesenchymal transition (EMT) and metastasis [9]. BNIP3 level could be regulated by oncogenes and subsequently influences the cell fate of different types of tumors, e.g., castration-resistant prostate cancer or pancreatic cancer [10,11]. Besides participating in cell death and metastasis-related processes, BNIP3 can control different metabolic pathways, such as lipid metabolism [12], glycolysis [13], and mitochondrial bioenergetics [14]. Hence, BNIP3 might act as a potential link between the death and survival of the cell [15]. The above-mentioned assets allow BNIP3 to have an impact on cancer development and progression. BNIP3 action is tissue-specific: it is differently expressed in normal cells and in different cancer types. In this review, we discuss the current data about BNIP3 involvement in lung cancer development and progression.

## 2. Characterization of BNIP3

The *BNIP3* gene is located on locus 10q26.3 and includes six exons. Its size is about 15 kilobase. The *BNIP3* promoter has a hypoxia response element (HRE). Under hypoxic conditions, hypoxia inducible factor 1α (HIF-1α) binds to the HRE and provokes *BNIP3* promoter activation. Consequently, it results in increased BNIP3 mRNA level sand its protein accumulation [16]. Von Hippel-Lindau protein defectiveness (e.g., in the RCC4 renal carcinoma cell line) results in constitutive HIF-1α expression and subsequent BNIP3 overexpression. BNIP3 is also upregulated in normoxic conditions and can be suppressed by HIF-1α silencing [8]. HIF-1α stabilization in normoxic conditions after treatment with CoCl_2_ and deferoxamine also induces BNIP3 expression [17].

The activity of the *BNIP3* promoter might also be silenced by methylation and histone deacetylation. These modifications occur in different types of cancer and may play a crucial role in their development. Thus, the *BNIP3* promoter is methylated in colorectal carcinoma [18], pancreas adenocarcinoma [19], several types of hematopoietic tumors [20] and BNIP3 re-expression by methyltransferase inhibitor restored hypoxia-inducible cell death. BNIP3 expression is suppressed by histone deacetylation in renal carcinoma [21] and hematopoietic tumors [20]. Inhibition of histone deacetylation also led to BNIP3 re-expression and cell death. Low BNIP3 levels might maintain cell death resistance and the survival of tumor cells.

The BNIP3 protein consists of 194 amino acids and has a predicted molecular weight of 21.5 kDa. BNIP3 presents as a monomer and as a homodimer. It includes four major domains (Figure 1).

First, the N-terminus domain contains the microtubule-associated protein 1A/1B-light chain 3 (LC3)-interacting region (LIR) motif. It ensures the targeting of BNIP3 to LC3 anchored in the membrane of the phagophore. BNIP3 homodimerization facilitates this interaction. The LIR-motif mediates the function of BNIP3 as an autophagy receptor [22]. The N-terminal region also contains PEST-sequences, which are rich in proline (P), glutamic acid (E), serine (S), and threonine (T). These sequences are flanked by histidine and arginine/lysine amino acid residues and are responsible for proteasome degradation [23].

The second domain is the Bcl-2 homology domain 3 (BH3-domain). Typically, the BH3-domain of pro-apoptotic Bcl-2 homologs mediates Bcl-2/Bcl-XL heterodimerization and confers pro-apoptotic activity. The BH3-domain of BNIP3 is atypical: it contains a tryptophan residue instead of aspartic or glutamic acid residue in the BH3-domain compared to other BH3-only members. Possibly, the tryptophan residue may reduce selectivity in interactions with pro-survival Bcl-2 proteins and weaken the pro-apoptotic function of BNIP3 [24]. There are studies confirming the stimulative effect of the BH3-domain of BNIP3 in cell death [25]. In contrast, there is an indication that the deletion of the BH3-domain does not abolish its pro-apoptotic ability, so this domain is not significant for cell death induction [26,27]. Therefore, the role of the BH3-domain in BNIP3 functioning is controversial. Moreover, the pro-apoptotic action of BNIP3 is weak in comparison to other BH3-only proteins.

The C-terminus domain fulfils transmembrane function and is responsible for mitochondrial BNIP3 localization and ensures its dimerization. This domain is essential for BNIP3 functioning as a cell death regulator [28]. BNIP3 lacking the transmembrane (TM) domain is not able to locate in mitochondria, dimerize and cause apoptosis [25,29]. Phosphorylation of this domain by survival kinases, such as protein kinase A, might inhibit the pro-death and mitochondrial degradation action of BNIP3 [30].

Mitochondrial BNIP3 localization via the C-terminal TM domain is crucial for its functioning. Deletion of this TM domain leads to abolishing of the heterodimerization of BNIP3 with pro-survival Bcl-2 proteins and BNIP3 pro-apoptotic activity [25]. Prevention of homodimerization does not inhibit the pro-death function of BNIP3 if its localization remains in the mitochondria [26,31]. Hence, these data claim that only subcellular localization (but not homodimerization) is essential for BNIP3-dependent apoptosis. In contrast, BNIP3-dependent autophagy is determined by both localization and homodimerization. BNIP3 mutants that lack mitochondrial localization and are characterized by disrupted homodimerization are not able to induce mitophagy. BNIP3 can also be localized on the membrane of the endoplasmic reticulum (ER), where it induces ER-phagy (autophagic ER removal) [22]. Therefore, not only mitochondrial but also ER localization of BNIP3 is critical for its functioning.

## 3. Role of BNIP3 in Cancer

BNIP3 is a stress sensor protein that is strongly induced by hypoxia [17]. As mentioned above, *BNIP3* containing HRE were shown to be a direct target of HIF-1α and implicated in hypoxia-induced cell death [15,32]. Originally, it was shown to be one of the mediators of hypoxia-related cell death in cardiac myocytes and neurons [33,34]. BNIP3 was found to play an important role in carcinogenesis [35]. The production of BNIP3 protein has been reported to be upregulated in lung, prostate, glioblastoma multiforme, cervical tumors, endometrial cancer, breast carcinomas, and gastric adenocarcinomas [36,37,38,39,40]. Furthermore, enhanced expression of BNIP3 is usually correlated with the aggressive tumor phenotype and a poor prognosis [35]. For instance, increased levels of BNIP3 protein were detected in NSCLC, where high expression of BNIP3 protein correlated with a poor prognosis in early stages [37]. Similarly, in prostate cancer, a significant correlation between cytoplasmic BNIP3 expression and unfavorable patient outcomes was found [41]. Furthermore, in this study BNIP3 overexpression was noticed to correlate with downregulation of tumor suppressor miR145 and repressed the *apoptosis-inducing factor* gene [41]. BNIP3 expression was shown to be increased in hypoxic regions of breast tumors [40,42]. Additionally, BNIP3 overexpression in ductal carcinoma was found to correlate with a high risk of tumor recurrence and shorter disease-free survival [43].

On the other hand, decreased levels of BNIP3 mRNA and protein were found in pancreatic and colorectal cancer, hematopoietic malignancies and hepatocellular carcinoma (HCC) [19,20,44,45]. Loss of BNIP3 expression contributed to resistance to therapy and a worse prognosis for the patient [46]. Indeed, in pancreatic adenocarcinomas, BNIP3 expression was repressed by hypermethylation of the *BNIP3* promoter contributing to cancer progression [19,44]. In turn, restoration of BNIP3 expression provided pancreatic cancer cell death [19]. Likewise, silencing of the *BNIP3* gene through aberrant methylation was correlated with resistance to therapy in colorectal cancer [47]. Epigenetic silencing of *BNIP3* was also found in HCC cells resistant to treatment [48]. Furthermore, more aggressive subtypes of HCC demonstrated a higher degree of *BNIP3* promoter methylation than less aggressive subtypes [38,46].

Thus, the role of BNIP3 in tumor progression and resistance to therapy remain highly context-dependent and could provide both tumor suppressing and tumor promoting mechanisms. The controversial role of BNIP3 in cancer could be explained by fact that its ability to induce cell death was inactivated [38]. For instance, Bcl-2 was shown to block BNIP3-induced cell death through binding to its TM domain [49]. In the breast cancer cell line, BNIP3-induced cell death was blocked by growth factor signaling [50]. Treatment with epidermal growth factor (EGF) and insulin-like growth factor (IGF) inhibited BNIP3-induced cell death under hypoxic conditions, whereas downregulation of EGF receptor signaling with antibodies against Erb2 provided an increase of hypoxia-induced cell death.

Furthermore, it is observed that BNIP3 does not localize to the mitochondria in several tumors. In several tumors, BNIP3 expression was sequestered to the nucleus, thereby blocking its apoptotic activity [51,52].

Further studies revealed that BNIP3 is involved in the regulation of autophagy and autophagic cell death [38]. Autophagy is an evolutionarily conserved mechanism involving the formation of autophagosomes that sequester cytoplasmic macromolecules and organelles before fusion with the endo/lysosomal compartment [53,54,55]. The role of autophagy in cancer is highly dependent on the type of tumor and its developmental stage. Activation or inactivation of autophagy can contribute differently to tumorigenesis. Reduced autophagy can contribute to tumor progression, whereas increased autophagy may be a mechanism for tumor survival under hypoxic, metabolic or therapeutic stress conditions [56]. BNIP3 was reported to contribute to autophagic cell survival under hypoxic conditions [53,57]. Indeed, ectopic expression of BNIP3 and Bcl-2/adenovirus E1B 19 kDa protein-interacting protein 3-like/BNIP3-like protein X (BNIP3L or NIX) induces autophagy in normoxia, whereas the ablation of BNIP3 with siRNA increased cell death in hypoxia [53]. BNIP3 could promote autophagy by several mechanisms [58]. First, by competing with Beclin-1 for binding of Bcl-2, thereby liberating Beclin-1 from Bcl-2 complexes that further lead to the activation of autophagy [53]. Another possible mechanism is the inhibition of Rheb, an upstream activator of mammalian target of rapamycin [58,59]. Besides cell survival regulation, BNIP3 has been reported to induce autophagic cell death under hypoxic conditions [32,60]. In malignant glioma cell lines, acute overexpression of BNIP3 induced cell death with autophagic traits under hypoxic conditions [60]. Furthermore, treatment with the autophagy inhibitor 3-methyladenine significantly reduced cell death induced by hypoxia, whereas caspase inhibitor z-VAD-fmk failed to repress cell death activation [60]. Similarly, arsenic trioxide (As_2_O_3_) has also been shown to increase BNIP3 expression, leading to autophagic cell death [61]. These results indicate that BNIP3 may induce both autophagy and autophagic cell death depending on the molecular context and tissue.

Recent evidence demonstrates that BNIP3 plays an important role in mitophagy and is associated with mitochondrial dysfunction [62,63]. Mitophagy is a specialized form of autophagy providing the degradation of damaged and dysfunctional mitochondria through interactions of key adaptor molecules at the outer mitochondrial membrane with processed LC3, a standard autophagosome marker on the membrane of the phagophore [64]. Mitophagy activation occurs through different pathways, including the phosphatase and tensin homolog (PTEN)-induced putative kinase 1 (PINK1)/Parkin pathway, which is triggered by membrane depolarization [65,66]. Another pathway is activated via HIF-1α through adapter molecules (i.e., BNIP3, NIX, and FUN14 domain containing 1 (FUNDC1)) [64]. However, in cancer cells, mitophagy frequently appears dysregulated [66]. Mutations in the mitophagy genes contribute to the inhibition of mitophagy and further lead to increased reactive oxygen species (ROS) production and accumulation of dysfunctional mitochondria, thus contributing to carcinogenesis. Similar to autophagy, mitophagy may provide both pro- and anti-tumorigenic actions based on the context [67,68]. For instance, in a mouse model of mammary tumors, BNIP3 acts as a tumor suppressor [69]. In this study, BNIP3 deletion promoted the tumor growth associated with reduced mitophagy, elevated ROS production, mitochondrial dysfunction, and activation of HIF-1α-dependent genes, including those promoting glycolysis, angiogenesis and metastasis [69]. Similarly, in breast cancer cell line MCF-7, IGF-1 signaling induces BNIP3-dependent mitophagy and turnover of mitochondria [70]. In turn, resistance to IGF-1 provided BNIP3 downregulation, reduced mitophagy and accumulation of dysfunctional mitochondria [70]. Additionally, BNIP3-induced mitophagy may play an important role in cancer cell survival after irradiation [71]. In head and neck squamous cell carcinoma, p53/BNIP3-mediated mitophagy was shown to limit glycolytic shift and maintain mitochondrial integrity. In contrast, p53 deletion in this tumor was associated with blocking of BNIP3-induced mitophagy metabolic shift and accumulation of abnormal mitochondria. Besides the tumor-suppressing role, BNIP3-induced mitophagy could stimulate cancer progression. Indeed, in melanoma, downregulation of BNIP3 led to a significant reduction of cell migration activity and vascular mimicry by remodeling the cytoskeletal actin configuration, attenuating the aggressive behavior of the tumor [72].

BNIP3 appeared to regulate both general autophagy and mitophagy through different mechanisms in context-dependent manner. The depletion of BNIP3 highly affects both processes [53,69]. However, further studies are required in order to completely elucidate the role of BNIP3 in autophagy and mitophagy modulation for development of therapeutic approaches in context of personalized medicine.

The controversial role of BNIP3 in cancer progression is not completely understood. One of the possible explanations could be provided by its alternative splicing via deletion of exon 3, resulting in a truncated splice variant with pro-survival properties that is dependent on pyruvate kinase isozyme M2 [13,71]. However, this statement is difficult to apply to all cancer types. Furthermore, implication of BNIP3 in cancer progression appeared to be tissue-specific and depends on the molecular context and signaling pathways involved.

## 4. BNIP3 in Lung Cancer

Based on The Human Protein Atlas database, BNIP3 overexpression has been reported in up to 80% of patients with lung cancer [73]. Similarly, high expression of BNIP3 is detected in 70% of patients with lung cancer subtypes, such as lung AD and SCC. Several research groups have also shown that BNIP3 overexpression is observed in the tumor tissues of about 57% of patients with NSCLC [74,75]. Despite the high frequency of BNIP3 overexpression in tumor specimens, according to 5-year survival, BNIP3 is not correlated with survival probability in patients with different stages of lung cancer and its subtypes, including lung AD and SCC [73]. However, in a cohort of patients with only early-stage NSCLC, high expression of BNIP3 is linked with poor prognosis [37,74]. Contradictions in the prognostic role of BNIP3 in different stages of lung cancer can be explained by the ambiguous action of autophagy and mitophagy, including those regulated by BNIP3. It can be assumed that BNIP3-dependent autophagic and mitophagic cell survival prevails in early-stage lung cancer, which results in the formation of a more aggressive lung tumor phenotype. In the later stages of lung cancer, isolated cell populations are formed, in which autophagy and mitophagy function simultaneously as factors for enhancing cell survival and death, thus leveling the prognostic role of BNIP3. It is worth mentioning that exclusively nuclear BNIP3 localization was detected in the samples of a subset of patients with NSCLC [37,75]. Nuclear localization of BNIP3 may exclude its contribution to the regulation of mitochondria-dependent cell death, provoking cancer progression. Giatromanolaki et al. have shown that nuclear localization predicts a poor prognosis in early-stage NSCLC [37]. However, a study involving a cohort of patients with different stages of cancer did not confirm the negative prognostic role of nuclear BNIP3 in patients with NSCLC [75]. Conflicting results may be explained by a small number of patients with lung cancer with exclusively nuclear BNIP3 localization, which reduces the possibility of reliable statistical analysis conclusions. Besides this, the mechanism of BNIP3 translocation to the nucleus is still unidentified, which limits our ability to reach unambiguous conclusions about the predictive role of nuclear BNIP3. BNIP3 was recently established as part of a group of biomarkers for predicting the survival of lung AD and SCC along with other five autophagy-related genes, such as *EIF4EBP1*, *TP63*, *ATIC*, *ERO1A*, and *FADD*. Multivariate analysis of the gene expression datasets from large-scale databases revealed the signature, including these autophagy-related genes, as effectors of the development and prognosis of patients with NSCLC with different stages [76]. The identification of this autophagy-related pattern provides clinicians an invaluable opportunity to determine high-risk types of lung AD and SCC and apply this knowledge for the development of targeted therapy.

Autophagy is triggered by hypoxia, which in turn increases BNIP3 expression [53]. In that way, BNIP3 is the link between hypoxia and autophagy. Indeed, co-expression of BNIP3 and HIF-1α is observed in clinical samples of patients with NSCLC [37]. This in vivo data correlates with the results obtained on NSCLC and SCLC cell lines, where hypoxia led to BNIP3 overexpression mediated by HIF-1,2α accumulation [77,78,79]. Moreover, hypoxia-induced HIF-1α results in BNIP3-dependent autophagy of NSCLC cells [77]. HIF-1α activation caused by 4-O-methylascochlorin (MAC), without stimulating hypoxia, also promotes BNIP3 overexpression. In leukemia cells, MAC augmented apoptotic cell death; however, in NSCLC cell lines, MAC activated autophagy and slightly decreased the cell survival rate [80]. The use of MAC on different cell lines shows the ambivalent link between cell death and BNIP3-dependent autophagy. Autophagy inhibition, including the BNIP3-mediated one, is a promising tool for increasing sensitivity to cell death by excluding pro-survival autophagic effects. As mentioned above, lung cancer development is associated with mutations in the *EGFR* gene. These facts have led to a combination therapy that involves the inhibitors of EGFR tyrosine kinase (lapatinib) and vacuolar-type ATPase (vATPase), including autophagy inhibitors (Bafilomycin A1 and Concanamycin A). The combination of these agents led to increased cell death by downregulation of HIF-1α and BNIP3 in NSCLC cells [81]. The pyrimidine antimetabolite, gemcitabine, has been approved by the FDA for cancer treatment; however, its prolonged use provokes cell death resistance. Gemcitabine causes BNIP3-related autophagy and leads to reduced lung cancer cell death, which may contribute to gemcitabine-acquired resistance [82]. In bladder cancer cells, suppression of BNIP3-regulated autophagy elevates gemcitabine-induced apoptosis and improves gemcitabine efficacy [83]. Arguably, the combination of gemcitabine and autophagy inhibitors may overcome gemcitabine-acquired resistance in lung cancer cells. Besides drug sensitivity and resistance, autophagy and BNIP3 participate in the response to ionizing radiation treatment. New autophagy inhibitor, Lys05, a dimeric form of chloroquine, decreased *BNIP3* gene expression and autophagy, which enhanced the radiosensitivity of NSCLC cells [84]. The development of chemical agents for BNIP3 targeting could be a prospective strategy to overcome radioresistance acquired during inoperable tumor therapy. In addition to autophagy, BNIP3 participates in cisplatin-induced cell death, including apoptosis [78,85]. Cisplatin treatment increases BNIP3 protein levels; however, its combination with hypoxia reduced BNIP3 in lung cancer cell lines. Simultaneous treatment of cisplatin with hypoxia stimulated autophagy decreased cell death. Thereby, a combination of cisplatin and hypoxia leads to elevated cell survival via switching on BNIP3-independent autophagy [78].

The histone deacetylases inhibitor, panobinostat, LBH589, is a promising agent for lung cancer therapy through the impairment of histone deacetylation and subsequent weakening of gene transcription, especially oncogenes. The combination of LBH589 and cisplatin triggers BNIP3 insertion into the mitochondrial membrane by its TM domain, which eventually promotes ROS and potential mitochondrial membrane loss, leading to apoptotic cell death in the lungs [85]. The molecular design of drugs for the stimulation of BNIP3 TM domain insertion into mitochondria could provide an efficient tool to enhance sensitivity to apoptosis. The involvement of BNIP3 in autophagy and control of mitochondrial status allows it to participate in energy-intensive processes, such as EMT and metastasis. BNIP3 is upregulated by the cell cycle protein p53. Mutual overexpression of p53 and BNIP3 leads to enhanced apoptosis in highly metastatic lung cancer cell lines compared to low metastatic ones. Yan et al. suggested that increased apoptosis in cells with greater mobility is due to the ability of BNIP3 to eliminate damaged cells. The exclusion of weak cells resulted in the formation of a cell population with high metastatic capacity [86]. Therefore, BNIP3 might act as a link between lung cancer cell death and its dissemination. BNIP3 is also regulated by the aryl hydrocarbon receptor (AhR) associated with autophagy and metabolism of xenobiotic compounds, including polycyclic aromatic hydrocarbons in cigarette smoke. Besides these functions, AhR has influences on cell migration and dissemination, but its impact is different and depends on tumor cell type [87]. Tsai et al. demonstrated that AhR augmented BNIP3 ubiquitination for proteasomal degradation, leading to autophagy suppression and decreased EMT of NSCLC cells in vivo and in vitro [88]. Hence, BNIP3 might be a link between the pathway of endogenous compounds metabolism and metastasis-related EMT of lung cancer cells. In addition to AhR, BNIP3 is controlled by the planar cell polarity effector protein fuzzy homolog (FUZ) in NSCLC cell lines. FUZ knockdown reduced BNIP3 protein levels but had no effect on the level of BNIP3 mRNA, which eliminates FUZ as an influencer of *BNIP3* gene expression. BNIP3 and FUZ are co-localized and interact with each other to retain their protein stability. FUZ stimulates EMT and promotes cell proliferation in lung cancer cell lines [89]. Moreover, FUZ is linked with poor prognosis in patients with NSCLC. The FUZ-BNIP3 axis may be considered as one of a potential group involving FUZ and BNIP3 as highly expressed proteins and FUZ as a prognostic factor of overall survival in patients with lung cancer.

## 5. Conclusions

BNIP3 is a crucial player in autophagy, apoptosis and EMT processes in cancer, in general, and in lung cancer, in particular (Figure 2). BNIP3 is frequently overexpressed during different stages of lung cancer and high BNIP3 expression was described as a major independent factor for overall NSCLC patients’ survival. Moreover, it was shown that the level of this protein has prognostic value in patients with early-stage NSCLC; however, its role in advanced NSCLC still must be clarified. BNIP3 was also established as one of the predictive factors in autophagy-related biomarker groups for lung AD and SC, suggesting that BNIP3 deserves additional investigation as a prognostic marker.

As mentioned above, besides NSCLC, lung cancer includes SCLC subgroup, which accounts significantly smaller cohort. Unfortunately, it is still little known about mitochondria involvement in SCLC development. Further study on SCLC will expand current knowledge concerning the development of different subtypes of lung cancer and, more precisely, the role of mitochondria and autophagy proteins, including BNIP3, in lung carcinogenesis.

BNIP3-dependent autophagy influences cell death sensitivity and resistance; therefore, targeting this type of death is a promising approach in cancer therapy. The association of BNIP3 with EMT-related proteins allows it to affect cancer cell dissemination and metastasis. Thus, keeping in mind all above-noted BNIP3 functions it make sense for much deeper research in this direction. Revealing of new BNIP3 mechanisms of action and its potential link to oncogenesis signaling could result in considering BNIP3 as an attractive candidate for targeted therapy of lung cancer in the near future. However, it is worth mentioning that radical inhibition of BNIP3 may decrease its levels in both normal and benign tumor tissues, resulting in ambiguous cell fate. In this respect, the design of low-molecular compounds and their delivery using nanocontainers may promote the development of BNIP3-targeted therapy.

## Figures and Tables

**Figure 1 cancers-12-03390-f001:**
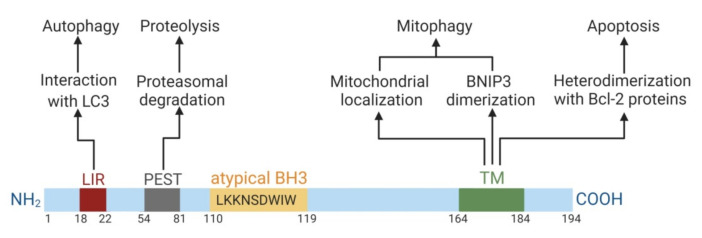
Domain structure of BNIP3 and its domain-dependent functions. Numbers correspond to amino acid counts. The illustration was created in BioRender.

**Figure 2 cancers-12-03390-f002:**
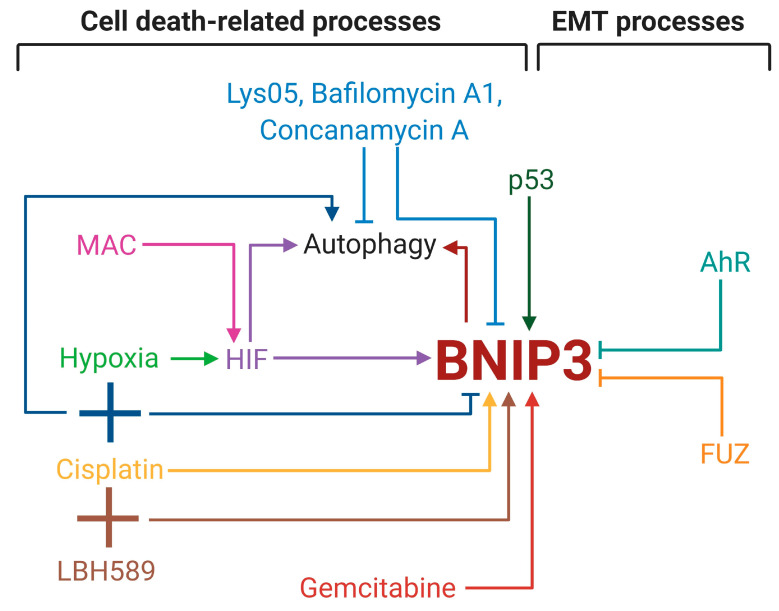
Influence of BNIP3 on cell death and epithelial-mesenchymal transition (EMT) processes in lung cancer. The illustration was created in BioRender.

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
