# Peer review of "BNIP3 in Lung Cancer: To Kill or Rescue?"

_cancers, 2020, doi:10.3390/cancers12113390_

Round 1

Reviewer 1 Report

I read with interest the review article by Gorbunova et al. entitled “BNIP3 in lung cancer: to kill or rescue?”. The authors explore the role of BNIP3 in several cancer metabolic pathways, with special focus on lung cancer.

The major concern of this reviewer about this article is that, though being an interesting reading, it seems that is not able to elucidate the effective role of such protein for lung cancer treatment. In fact, through the entire review the authors explain the controversy and not well elucidated role of BNIP3 in various cancers, including lung cancer, and the conclusion that it could be a targetable protein for LC treatment seems not to be adequate. Moreover, BNIP3 was established to play a prognostic role only in early-stage NSCLC, while its role in advanced NSCLC still has to be clarified.

Minor concerns:

  • Being a review article, there is no need to organize the abstract as background, methods, results, as this refers to original articles, or to meta-analyses if at all.
  • Line 44: Squamous cell carcinoma (SCC), large cell carcinoma (LCC) and adenocarcinoma (AD) are the most represented subtypes of non-small cell lung cancer (NSCLC). The authors should rephrase, as the sentence supposes there are only 3 NSCLC subtypes.
  • Line 47: as the authors mention NSCLC activating mutations and TP53, it would be fashionable to insert the relationship related to prognosis of this mutation (PMID 32272775).
  • Line 47: as a percentage of NSCLC is oncogene-addicted, the authors could add some information about the possible role of BNIP3 and activating mutations patterns, as done in other pathologies (PMIDs 32626951; 32565538).
  • Figures 1 and 2: there is no need to insert a phrase resending to text for details.
  • Line 316: does the authors intend low-molecular weight molecules?

Reviewer 2 Report

This review article by Gorbunova et al, focuses on the role of BNIP3, a peculiar member of the BH3-only protein of the Bcl-2 family, in lung cancer. This is the first time that the implication of this singular protein has been compiled in one review in cancer context. The manuscript is overall very well written and presents a synthetic view showing essential points.

Based on this, I have minor concerns :

I would recommend the authors to improve the iconography by including a figure compiling the principal pathways controlled by BNIP3, linked with the different domains of the protein.

The role of Bnip3 in autophagy process is well known. However, Bnip3 can participate in the global autophagy but also in more selective pathway, like mitophagy. The authors may explain more precisely if the autophagy control by Bnip3 is more related to one of the other (or both).

Finally, the authors decided to highlight the role of Bnip3 in lung cancer, but without arguing why lung cancer may be more relevant than other type of cancer to focus on the role of Bnip3 and their potential targeting.

Round 2

Reviewer 1 Report

The authors effectively clarified the issues pointed out by this reviewer. No other issues are requested by this reviewer, that considers the present paper ready for publication.